# Potential plasma biomarkers: miRNA-29c, miRNA-21, and miRNA-155 in clinical progression of Hepatocellular Carcinoma patients

**Neneng Ratnasari**[1]*, **Puji Lestari**[2☯¤a], **Dede Renovaldi**[2☯¤b], **Juwita Raditya Ningsih**[2☯¤c], **Nanda Qoriansas**[2☯], **Tirta Wardana**[3☯], **Suharno Hakim**[4‡], **Nur Signa Aini Gumilas**[5‡], **Fahmi Indrarti**[1‡], **Catharina Triwikatmani**[1‡], **Putut Bayupurnama**[1‡], **Didik Setyo Heriyanto**[6‡], **Indwiani Astuti**[7‡], **Sofia Mubarika Harjana**[8‡]

1 Gastroenterology-Hepatology Division of Internal Medicine, Department Faculty of Medicine, Public Health and Nursing Universitas Gadjah Mada/ Dr. Sardjito General Hospital, Yogyakarta, Daerah Istimewa Yogyakarta, Indonesia, 2 Graduate School of Biotechnology Universitas Gadjah Mada, Daerah Istimewa Yogyakarta, Indonesia, 3 Department Biomedicine, School of Dentistry, Faculty of Medicine Jenderal Soedirman University, Jawa Tengah, Indonesia, 4 Internal Medicine Department Dr. Margono Soekarjo Hospital/Faculty of Medicine Universitas Jendral Soedirman, Jawa Tengah, Indonesia, 5 Histology Department Faculty of Medicine Universitas Jendral Soedirman, Jawa Tengah, Indonesia, 6 Anatomic Pathology Department, Faculty of Medicine, Public Health and Nursing, Universitas Gadjah Mada, Yogyakarta, Daerah Istimewa Yogyakarta, Indonesia, 7 Pharmacology and Therapy Department, Faculty of Medicine, Public Health and Nursing, Universitas Gadjah Mada, Yogyakarta, Daerah Istimewa Yogyakarta, Indonesia, 8 Histology and Cell Biology Department, Faculty of Medicine, Public Health and Nursing, Universitas Gadjah Mada, Yogyakarta, Daerah Istimewa Yogyakarta, Indonesia

☯ These authors contributed equally to this work.
¤a Current address: Department of Research & Development, Dharmais Cancer Hospital, Jakarta Barat, Indonesia
¤b Current address: Department of Biomedical Sciences, Faculty of Medicine and Health, Universitas Muhammadiyah Jakarta, Banten, Indonesia
¤c Current address: Faculty of Dentistry, Universitas Muhammadiyah Surakarta, Jawa Tengah, Indonesia
‡ SH, NSAG, FI, CT, PB, DSH, IA and SMH also contributed equally to this work.
* nenengratnasari@gmail.com

**Data Availability Statement:** All relevant data are within the manuscript.

## Abstract

This study evaluated differences in the clinical appearance of patients with hepatocellular carcinoma (HCC) based on plasma level and regulation of microRNAs (miRNA-29c, miRNA-21, and miRNA-155). The observational-analytical study with a cross-sectional design was conducted on 36 HCC patients and 36 healthy controls. The blood samples were collected from 2 Province Hospitals (Dr. Sardjito Hospital and Prof. Dr. Margono Soe-karjo Hospital) for HCC and the Blood Bank Donor of the Indonesian Red Cross for 36 healthy controls. These blood samples were treated as follows: plasma isolation, RNA isolation, cDNA synthesis, quantification by qRT-PCR using a sequence-specific forward primer, and normalization of miRNA using housekeeping-stably miRNA-16. There were only 27 HCC patients with complete clinical variables (neutrophil to lymphocyte ratio (NLR), platelet to lymphocyte ratio (PLR), platelet count, albumin, C-reactive protein (CRP), and cholines-terase (ChE)) that were able to analyses for regulation miRNAs based on its fold change expression miRNA target. All 27 HCC subjects were follow-up until 3-years of monitoring for

**Funding:** The author(s) received no specific funding for this work.

**Competing interests:** The authors have declared that no competing interests exist.

their overall survival. The miRNA plasma expression was analyzed by Bio-Rad CFX 96 Manager software to determine the cycle of quantification, followed by the calculation of expression levels using Livak's methods. Data were analyzed using STATA 11.0, with a significant value of p<0.05. The miRNAs expression of HCC subjects were lower than that healthy control subjects in miRNA-29c (down-regulation 1.83-fold), higher than that healthy control subjects in miRNA 21 and miRNA-155 (up-regulation, 1.74-fold; 1.55-fold) respectively. NLR, CRP, ChE, and platelet count showed a significant difference in miRNA-29c regulation, though neutrophil count showed a significant difference in miRNA-21 and miRNA-155 regulation (p<0.05). Conclusion: Plasma biomarkers: miRNA-21 and miRNA-155 might be potential biomarkers as onco-miR in HCC subjects, while miRNA-29c might act as a tumor suppressor. Significant evidence was identified with clinical progression based on the regulation of miRNAs, which was consistent with miRNA -29c.

## Introduction

Hepatocellular carcinoma (HCC) is the fifth most common of the major malignant liver tumors in the world and the third leading cause of mortality-related cancer. The HCC incidence is varied worldwide based on geographic variation, with 81% of HCC cases occurring in developing countries and 54% of those from China [1, 2]. The majority of primary HCC (70–80%) which are mostly rare are detected at early stages, with the consequence of a decline of the mortality rate [1, 2].

More than 90% of HCC cases develop in the chronically inflamed liver as a result of viral hepatitis, alcohol abuse, and increased incidence in patients with non-alcohol fatty liver disease. Most of the HCC cases in developing countries are caused by hepatitis B virus (HBV) (58.8%) and hepatitis C virus (HCV) (33.4%), with different prevalence, compared with developed countries with hepatitis B virus (23.3%) and hepatitis C virus (19.9%) [3, 4]. Prediction of cancer progression was revealed as 5.3–148 times higher in HBsAg positive and 20 times higher in anti-HCV positive [3, 4].

There is various pathogenesis of HCC development in both molecular and histological views. The alteration of essential gene regulation of cellular processes has a role in these processes, such as cell cycle control, cell growth, apoptosis, migration, and cell deployment. MicroRNAs (miRNAs) are one of the protein regulators that are associated with the carcinogenesis pathway. The involvement of miRNAs in tumorigenesis and tumor progression is well established because they can act as a tumor suppressor or promoter of oncogenesis depending on the cellular function of their targets. So far, 20–80% of transcribed human genes are regulated by miRNAs which are present not only in tissue cells but also in body fluids [5–7].

The dysregulation of miRNAs may affect some crucial biological processes of cells leading to tumor development by increasing proliferation, decreasing apoptosis, and enhancing the metastatic potential. The miRNA-122 is the most abundant miRNA in the normal liver parenchyma, accounting for more than 70% of the total miRNA in hepatocytes [8]. The miRNA-21 is one of the first detected oncogenic miRNAs, controlling the cell cycle and tumorigenesis. The serum level of miRNA-21 was significantly higher than miRNA-155 and miRNA-365 in patients with breast cancer compared to healthy subjects. Plasma levels of miRNA-122 and miRNA-21 have strong correlations with the degree of fibrosis in HCV genotype-4 patients. Consequently, they can be considered as a potential biomarker for the early detection of

hepatic fibrosis. Moreover, miRNA-21 can be used as a potential biomarker for early detection of HCC combined with vascular endothelial growth factor (VEGF) and α-Fetoprotein (AFP) [9–11]. And also, the serum level of miRNA-21 is able for detecting the liver necroinflammatory activity event there was not the presence of HCC [12].

Previous studies mentioned that miRNA-21 acts as an onco-miR, while miRNA-29c and miRNA-155 act as tumor suppressor miRNA in HCC. The three miRNAs might be detected in HCC and could be used as minimally invasive biomarkers in HCC detection. The miRNA-155 expression levels showed significant elevation in HCC tissue with an increase of 1.5–6 times compared to normal liver tissue [13, 14].

Alanine aminotransferase (ALT) serum level remains a more common feature as a biomarker for liver injury dysfunction more than liver functions only. The serum AFP is the most widely used tumor biomarker currently in HCC patients even though the specificity of AFP is low [15]. However, AFP is usually used as a clinical biomarker of severity of liver inflammation in patients with HCC and screening of HCC evidence [16, 17].

Previous studies reported about the inflammation indexes that are useful for determining mortality and morbidity risk in HCC and Liver Cirrhosis (LC), such as neutrophil-lymphocyte ratio (NLR) and platelet-lymphocyte ratio (PLR). Elevated NLR is positively associated with the pathogenesis and progression of primary HCC caused by HBV. Elevated PLR indicated a poor prognosis for patients with primary HCC. PLR may be a reliable, easily obtained, and low-cost biomarker with prognostic potential for primary HCC [18–20].

Cholinesterase (ChE) is an enzyme that is synthesized in the hepatocytes and released into the blood as acetylcholinesterase in red blood cells and butyric cholinesterase in serum. Activity and synthesis of ChE would be reduced if there is any cellular liver dysfunction following with damaging of the cell membrane and this is a cause of diminished reserve function. A previous study revealed that the alteration of ChE in the cirrhotic liver may reflect the changes in the pathophysiology of synthesis at the protein and miRNA levels, even though enzymatic activity does not change [21]. The blood level of ChE was significantly reduced in LC and HCC compared with chronic hepatitis C and healthy livers [22]. A combination of ChE blood level with severity disease using Child-Pugh score can be used as a reliable measurement of the liver reserve function [23].

C-reactive protein (CRP) is one of the inflammation biomarkers which are produced in the hepatocytes. High levels of CRP reflect an inflammation process in patients with infection and also in reactive inflammation in ongoing tumor growth. Inflammatory conditions may stimulate angiogenesis and promote invasive tumor progression. CRP level can be used as a prognostic marker in LC and HCC patients [24–27]. A combination of inflammation biomarkers (CRP and NLR) can be used as a predictor of tumor response and survival [28].

## Materials and methods

### Study population and data collection

The study protocol was approved by Medical and Health Research Ethics Committee (MHREC) Faculty of Medicine Gadjah Mada University—Dr. Sardjito General Hospital. The approval letter number: KE/FK/625/EC/2016. All participant read the research protocol and signed an informed consent before recruited in the study. All participants must meet the inclusion and exclusion criteria of the study.

This research was an analytical, observational study with a cross-sectional design conducted on 36 HCC patients and 36 healthy controls. Subjects were recruited from two General Hospitals in Central Java, Indonesia. The inclusion criteria were age 18–65 years old, primary HCC diagnosis was confirmed by histology examination, and two imaging (Doppler ultrasound and

3 phase MSCT-scan abdomen). The exclusion criteria were severe infectious diseases, cardio-vascular disease, and cancers originating from other organs. All subjects who met the criteria of study read and signed the informed consent form before the study began. There were 27 HCC subjects with complete clinical data continued for the clinical study and variable analysis of the relationship between subject characteristics with the regulation of miRNAs. All 27 HCC subjects were follow-up until 3-years of monitoring for their overall survival. The 36 healthy controls were recruited from the Blood Bank Donor list of the Indonesian Red Cross.

Estimation of sample size was calculated using online sample size calculation (www.raosoft.com/samplesize.html) based on hospital rate of HCC patients during 2010–2015 which was 60 patients per year, 5% margin error, 90% confidence level, and 50% response distribution.

## Micro-RNA analyses

**RNA extraction.**   The RNA isolation was prepared from blood plasma (6 mL blood venous was fixed by EDTA, centrifuged at 1500 rpm 10 minutes). The supernatant was transferred to a 1.5 ml micro scaping tube and then the sample code was then stored at -80°C. Total RNA isolation was performed from 200 µl of blood plasma using miRCURY RNA Isolation Kit-Biofluid (Cat No.300112, Exiqon) according to the kit procedure.

**cDNA synthesis.**   The total RNA isolation for miRNAs was performed using miRCURY RNA isolation Kit-Biofluid, followed by cDNA synthesis for miRNA using Universal cDNA synthesis kit ll, 8–64 rxns (Cat No.203301, Exiqon) and thermal cycler PCR (Biorad c 1000). The master mix consists of 4 µL 5x reaction buffers, 9 µL Nuclease free water, 2 µL Enzyme mix, and 1 µL Spike in (sp6), up to a total volume of 16 µL reagents. 4 µL RNA samples were added to the tube. Thermal cycler Biorad C1000 was used with the protocol 60 min at 42°C for incubation time, 5 min at 95°C for inactivation of reverse transcriptase enzyme and cooled at 4°C.

**Real-time Quantitative PCR.**   Real-time Quantitative PCR (qRT-PCR) for miRNAs was performed using Real-time qPCR (Biorad CFX 96). The sequence-specific primers were used for every miRNA target: miRNA 29c-3p, miRNA 21-5p, and miRNA 155-5p. The expression of every miRNA was normalized to housekeeping- stably expressed of miRNA-16. The sequence-specific forward primers based on miRbase (https://www.mirbase.org/cgi-bin/) were:

5′ UAGCACCAUUUGAAAUCGGUUA 3′ for miRNA 29c-3p;

5′-UAGCUUAUCAGACUGAUGUUGA-3′ for miRNA 21-5p;

5′-UUAAUGCUAAUCGUGUAGGGU-3′ for miRNA 155-5p; and

5′-UAGCAGCACGUAAAUAUU GGCG-3′ for miRNA-16.

The reagent for performing real-time qPCR miRNA was ExiLent SYBR Green master mix, 2.5 mL (Cat No. 203402, Exiqon), all primers set (miRNA-29c-3p, miRNA-21-5p, miR-155-5p, and cDNAs) prepared before. Reaction compositions consist of 5 µl SYBR Green master mix, 1µl PCR primer mix, 4µl cDNA sample which has been diluted with a ratio of 1:40.

Real time qPCR program Biorad CFX 96 were set up as follows: Denaturation 95°C for 10 min, amplification 40 cycle, at 95°C for 10 sec, 58°C for 1 min ramp-rate 1,6°C/s optical read and melt curve analysis.

The relative expression of the target gene was calculated by the cycle of quantification (Cq) analysis using the Livak's method [29]. The Relative expressions of miRNA formula:

$$\Delta Cq_{(miRNA\ target.HCC)} = Cq_{(miRNA\ target\ HCC)} - Cq_{(mirRNA-16\ HCC)}$$

$$\Delta Cq_{(miRNA\ target,healthy)} = Cq_{(miRNA\ target,\ healthy)} - Cq_{(miRNA-16,\ healthy)}$$

$$\Delta\Delta Cq_{(\text{miRNA target})} = \Delta Cq_{(\text{miRNA target, HCC})} - \Delta Cq_{(\text{miRNA target, healthy})}$$

$$\text{Fold change expression miRNA target} = 2^{-\Delta\Delta Cq}$$

Based on the result of fold change expression for every miRNA, all miRNAs in the HCC sample have adjusted the regulation to be up-regulation or down-regulation. The sample storage, RNA isolation, and cDNA synthesis were done at the Integrated Research Laboratory, Radiopoetro Building, Faculty of Medicine Public Health and Nursing Universitas Gadjah Mada. The qRT-PCR analysis was done at the Genetic Engineering Laboratory of the Center for Biotechnology Studies Universitas Gadjah Mada.

### Clinical laboratory examination

The automatic machine with a flow-cytometer was used for blood count examination. Plasma CRP was examined using the agglutination method or sandwich immunometric method. Plasma ChE was examined using colorimetric analysis. Plasma albumin was examined using a comprehensive metabolic panel. The clinical laboratory examination was conducted at the Clinical Pathology Laboratory of Dr. Sardjito General Hospital, Yogyakarta, Indonesia.

### Statistical analysis

The expressions of miRNA 29c-3p, miRNA 21-5p, and miRNA 155-5p were analyzed by Bio-Rad CFX Manager 96 Software to obtain the cycle of quantification (Cq), melting curve, and melt peak curve values. The differences in the expression of miRNAs between the blood plasma of patients with HCC and healthy individuals were tested using the Mann-Whitney test. The difference in clinical variables based on the fold change expression of every mRNAs ($2^{-\Delta\Delta Cq}$) on 27 HCC patients was calculated using the Mann-Whitney test. Based on the cut-off regulation of every miRNA target, how far the clinical predictors may predict the regulation of miRNAs was analyzed by Odds Ratio (OR), with $p < 0.05$ as a significant value. The survival analyses were calculated using Kaplan Meier analyses. Data analysis was calculated using STATA 11.0.

## Results

### The miRNAs expression level and its regulation in HCC subject and healthy control

All 36 HCC subjects who enrolled in the study continued to examine the miRNAs expression levels ($\Delta$miRNA) for miRNA 29c-3p, miRNA 21-5p, and miRNA 155-5p [11, 14]. The sequence-specific forward primers were used for the selection of the miRNAs targeted and miRNA-16 was used as a control (housekeeping- stably) miRNA [12]. Based on gender distribution, there were more males (23 patients; 63.9%) than females (13 patients; 36.1%), and the median age was 53 years old (min 29; max 74). The 36 healthy control subjects have recruited the study for knowing the control expression of miRNAs levels ($\Delta Cq_{\text{miRNAs}}$). The $\Delta Cq_{\text{miRNA}}$ $_{155\text{-}5p}$ and $\Delta Cq_{\text{miRNA 21-5p}}$ in HCC patients were lower than healthy controls, but only $\Delta Cq_{\text{miRNA 155-5p}}$ showed significantly different values with $p < 0.05$. The $\Delta Cq_{\text{miRNA 29c-3p}}$ in HCC patients was higher than healthy controls, even though the result was not statistically significant ($p > 0.05$) (Table 1).

Based on Livak's method, the proportion of down-regulation and up-regulation miRNAs were analyzed based on the fold change ($2^{-\Delta\Delta Cq}$) [29]. It showed that increasing regulations in miRNA 21-5p and miRNA 155-5p (1.74 and 1.55 folds); while in miRNA 29c-3p, there was

**Table 1. The difference in ΔCq, ΔΔCq and fold change of miRNA 29c-3p, miRNA 21-5p, miRNA 155-5p in HCC patients compared with healthy controls.**

| Variables* | | HCC (36) | Healthy (36) | P# | Fold change$ (2^{-ΔΔCq}) | Fold Regulation | Regulation in HCC¥ | |
|---|---|---|---|---|---|---|---|---|
| | | | | | | | down (%) | up (%) |
| mRNA 29c-3p | ΔCq miRNA 29c-3p | 6.96 ± 2.63 | 6.09 ± 1.83 | 0.180 | 0.6 | *Decreasing regulation 1.83 folds* | 19 (52.78) | 17 (47.22) |
| | ΔΔCq miRNA 29c-3p | 0.795 | | | | | | |
| miRNA 21-5p | ΔCq miRNA 21-5p | 0.43 ± 2.23 | 1.23 ± 2.16 | 0.126 | 18.73 | *Increasing regulation 1.74 folds* | 13 (36.11) | 23 (63.8) 9) |
| | ΔΔCq miRNA 21-5p | -3.28 | | | | | | |
| miRNA 155-5p | ΔCq miRNA 155--5p | 7.11 ± 1.45 | 7.74 ± 0.94 | 0.03 | 1.55 | *Increasing regulation 1.55 folds* | 10 (27.78) | 26 (72.22) |
| | ΔΔCq miRNA 155-5p | -0.631 | | | | | | |

The expression of every miRNA was normalized to housekeeping-stably expressed of miRNA-16; HCC: hepatocellular carcinoma

*) analyzed by Biorad CFX 96 Monogerm;

#) analyzed by Mann-Whitney with significant value p<0.05;

$)Livak's Method;

¥) calculated based on the value of fold change;

HCC: Hepatocellular carcinoma

decreasing regulation of 1.83-fold (Table 1). Based on the fold change analysis, miRNA 29c-3p may be acting as a tumor suppressor, and oppositely, miRNA 21-5p and miRNA 155-5p may act as onco-miR [12, 13].

## The relationship between clinical characteristics and miRNA regulation in HCC subjects

Subjects who had complete based line clinical data and the ΔCq level of miRNA were continuing statistical analyzes. Only 27 HCC patients had completed data.

The characteristic data and clinical laboratory examination showed in Table 2, showing 62.96% of subjects were male with a median of age 55 years old. The increase of AFP ≥200 IU/

**Table 2. The characteristic data for subjects with HCC (27 patients).**

| Variables | N (%) | Variables | N (%) | Median (min; max) |
|---|---|---|---|---|
| Sex | | Age (i.e.) | - | 55 (29;74) |
| • Male | 17 (62.96) | | | |
| • Female | 10 (37.04) | | | |
| Alpha-fetoprotein (≥ 200 IU/mL) | 13 (48.15) | Cirrhosis | 18 (66.67) | - |
| Hepatitis | | BCLC stage | | - |
| • HBV | 16 (59.26) | • B | 8 (29.63) | |
| • HCV | 2 (7.41) | • C | 15 (55.56) | |
| • None | 9 (33.33) | • D | 4 (14.81) | |
| Nodule characteristics | | Treatment | | - |
| • Multiple | 19 (70.37) | • TACE | 14 (51.85) | |
| • Diameter ≥ 5 cm | 23 (85.18) | • Sorafenib | 3 (11.11) | |
| • Metastasis | 5 (18.52) | | | |
| • Thrombus portal vein | 2 (8.33) | | | |
| Histories | | | | |
| • Smoking | 7 (25.93) | | | |
| • Alcohol | 3 (11.54) | | | |

Min: minimum; max: maximum; yon: year old; BCLC: Barcelona Clinic Liver Cancer; TACE: transarterial chemoembolization; HCC: Hepatocellular carcinoma

mL was around 48.15%, with most of the subjects suffering from HBV infection (59.26%) and cirrhosis background (66.67%). 85.18% of subjects were within large tumors >5 cm in diameter and others with multiple tumors (70.37%). The distribution of BCLC staging C showed that more frequent than B and D. There were only 17 patients who underwent chemotherapy. Trans-arterial chemoembolization (TACE) underwent for 14 patients and sorafenib for three patients only.

The clinical laboratory results showed that the mean of neutrophil, platelet, lymphocyte, and choline esterase was within the normal limit. The slight decrease of mean albumin and increase of mean CRP was shown in the laboratory results in Table 3. There was increasing in the mean of NLR (5.94 ± 3.62) based on a normal range [1–3], and there was a normal mean PLR based on the median normal range (132.4 ± 43.68) [30].

The difference of miRNAs regulation based on clinical laboratory data showed a significant difference of miRNA 29c-3p regulation with NLR and platelet count (value in down-regulation higher than up-regulation, p<0.05), and ChE level (value in down-regulation lower than up-regulation, p<0.05). Increasing in NLR and platelet; decreasing of ChE and albumin showed a significant difference in miRNA 29c-3p regulation. Although, in miRNA21-5p and miRNA 155-5p, there was only an increasing neutrophil that showed a significant difference (Table 4).

The OR analyses between miRNAs regulation and clinical variables showed in Table 5. Using cut-off value of NLR ≥4 (AUC = 0.7765, sensitivity 76.47%, and specificity 80% respectively; Fig 1A). It means there is a 77.65% chance that the model will be good to distinguish between positive class and negative class. NLR may be a predictive factor of up-regulation in miRNA 21-5p (OR = 3) and miRNA 155-5p (OR = 2.63), and even as a protective factor in miRNA 29c-3p (OR = 0.11). NLR with cut-off≥ 4 was consistent as clinical predictive of miRNA 21-5p and miRNA 155-5p, even as a preventive factor in miRNA 29c-3p.

Using cut-off value of PLR ≥165.83 (AUC = 0.5882; 58.82% sensitivity, and 40% specificity respectively; Fig 1B). It means there is a 58.82% chance that the model will be unsatisfactory to distinguish between positive class and negative class. PLR may be a predictive factor up-regulation of miRNA 29c-3p (OR = 1.6) and protective factor up-regulation of miRNA 21-5p (OR = 0.65). However, if the cut-off value of PLR was used the tertile 1 (T1) (PLR ≥ 292.40) [31], the quality of the PLR cut-off was better than the AUC result. PLR may be a predictive factor up-regulation of miRNA 21-5p (OR = 1.62) and miRNA 155-5p (OR = 3.11), even PLR in miRNA 29c-3p (OR = 0.25) may be changed to be a protective factor. The cut-off PLR using tertile 1 (T1≥ 292.40) was more consistent than PLR ≥165.83 (cut-off based on AUC analyses) in miRNAs regulation.

**Table 3. The clinical laboratory results (27 HCC patients).**

| Variables | Mean ± SD | Median (min; max) |
|---|---|---|
| Neutrophil ($10^3$/μL) | 6.76 ± 2.76 | 6.1 (3.37; 15.41) |
| Lymphocyte ($10^3$/μL) | 1.31 ± 0.51 | 1.2 (0.46; 2.61) |
| Platelet ($10^3$/μL) | 270.15 ± 111.11 | 243 (118; 722) |
| Albumin (g/dL) | 3.36 ± 0.66 | 3.42 (2.16; 4.4) |
| Choline esterase (U/mL) | 13.26 ± 18.42 | 6.7 (2.8; 77) |
| C-reactive protein (mg/L) | 44.46 ± 39.40 | 41 (5; 131) |
| Neutrophil-Lymphocyte ratio | 5.94 ± 3.62 | 4.11 (2.32; 15.1) |
| Platelet-Lymphocyte ratio | 234.94 ± 137.71 | 196.03 (91.47; 694.23) |

SD: standard deviation; min: minimum; max: maximum; HCC: Hepatocellular carcinoma

**Table 4. The regulation of miRNA analyses based on clinical variables in 27 HCC patients.**

| Variables | miRNA 29c-3p (n) | | | miRNA 21-5p (n) | | | miRNA 155-5p (n) | | |
|---|---|---|---|---|---|---|---|---|---|
| | down-reg[#] (9) | up-reg (18) | p* | down-reg (13) | up-reg[##] (14) | p* | down-reg (10) | up-reg[##] (17) | p* |
| NLR | 6.84±4.01 | 4.12±1.69 | *0.015* | 6.48±3.90 | 5.44±3.42 | 0.331 | 7.84±4.64 | 4.82±2.37 | 0.175 |
| PLR | 245.53±155.73 | 213.78±96.63 | 0.151 | 244.46±153.32 | 226.1±126.70 | 0.662 | 253.51±108.81 | 224.02±154.32 | 0.228 |
| CRP | 45.55± 41.30 | 42±40.62 | 0.974 | 50.62±41.05 | 34.6±38.87 | 0.462 | 67.33±55.52 | 37.6±34.04 | 0.308 |
| Platelet | 274.95±127.73 | 256.8±58.29 | *0.018* | 311.31±140.22 | 234.0±56.37 | 0.147 | 286.8±71.36 | 259.16±123.73 | 0.089 |
| ChE | 8.98 ±6.16 | 21.72±30.97 | *<0.001* | 10.25±6.75 | 15.67±24.89 | 0.834 | 8.8±7.05 | 13.92±19.60 | 0.840 |
| Lymphocyte | 1.29±0.54 | 1.36±0.47 | 0.670 | 1.44±0.58 | 1.20±0.42 | 0.497 | 1.29±0.59 | 1.33±0.48 | 0.422 |
| Neutrophil | 7.52±2.93 | 5.22±1.60 | 0.075 | 8.03±3.04 | 5.57±1.87 | *0.017* | 8.47±3.46 | 5.75±1.64 | *0.021* |
| Albumin | 3.29±0.76 | 3.49±0.24 | *0.002* | 3.20±0.74 | 3.49±0.52 | 0.153 | 3.29±0.66 | 3.40±0.64 | 0.792 |

NLR: Neutrophil-Lymphocyte ratio; PLR: Platelet-Lymphocyte ratio; CRP: C-Reactive protein; ChE: Choline esterase; down-reg: down regulation; up-reg: up regulation; HCC: Hepatocellular carcinoma; reg: regulation

*) Significant value of *p*<0.05; using Mann-Whitney test

#) tumor suppressor;

##) oncomiR

The minimal normal range of ChE (<7) used as a cut-off value, ChE may be a preventive factor of miRNA 29c-3p and miRNA 21-5p up-regulation (OR = 0.44; OR = 0.75), even as a predictive factor of miRNA 155-5p up-regulation (OR = 2). Based on cut-off albumin value <3.2 showed that albumin might be a predictive factor up-regulation of miRNA 21-5p (OR = 3.89), even though as preventive factor up-regulation in miRNA 155-5p (OR = 0.55) and miRNA 29c-3p (OR = 0.14).

Based on APASL criteria of AFP ≥200 IU/ml, this study results showed AFP might be a predictive factor (OR = 1.25) of up-regulation of miRNA 29c-3p and preventive factor for up-regulation of miRNA 21-5p (OR = 0.32) and miRNA 155-5p (OR = 0.42).

Based on nodule characteristics, multiple nodules and nodules more than 5 cm in diameter were showed no significant differences for every miRNA's regulation (p>0.05). Likewise, BCLC staging-C is more dominant than BCLC-B and BCLC-D, but no significant difference for every of miRNA's (p>0.05) (Table 6).

Hepatitis B virus was more frequent etiology compared with hepatitis C virus and other etiology. Significant difference was showed in miRNA 21-5p (p = 0.009) and miRNA 155-5p

**Table 5. The odds ratio analyses of up-regulation miRNA based on cut off value clinical variables in 27 HCC patients.**

| Variables | miRNA 29c-3p[#] | | | miRNA 21-5p[##] | | | miRNA 155-5p[##] | | |
|---|---|---|---|---|---|---|---|---|---|
| | OR | min; max | p | OR | min; max | p | OR | min; max | p |
| NLR≥4 | 0.11 | 0.01; 0.92 | *0.014* | 3 | 0.49;19.82 | 0.168 | 2.63 | 0.4;20.54 | 0.247 |
| PLR ≥165.83 | 1.6 | 0.24;12.89 | 0.580 | 0.65 | 1.10;3.90 | 0.581 | 1.05 | 0.17;7.11 | 0.952 |
| PLR (T1) ≥ 292.40 | 0.25 | 0.004; 2.85 | 0.214 | 1.62 | 0.21;13.96 | 0.580 | 3.11 | 0.37;27.21 | 0.201 |
| ChE< 7 | 0.44 | 0.03; 6.24 | 0.464 | 0.75 | 0.06;8.58 | 0.595 | 2 | 0.08;137.35 | 0.605 |
| Alb<3.2 | 0.14 | 0.00; 1.61 | 0.070 | 3.89 | 0.56;31.12 | 0.107 | 0.55 | 0.07;3.75 | 0.483 |
| AFP≥200 | 0.8 | 0.12;5.34 | 0.785 | 0.48 | 0.08;2.77 | 0.332 | 0.30 | 0.04;2.01 | 0.148 |

NLR: Neutrophil-Lymphocyte ratio; PLR: Platelet-Lymphocyte ratio; PLR (T1): Platelet-Lymphocyte ratio, first tertile value; CRP: C-Reactive protein; ChE: Choline esterase; Alb: albumin; AFP: alpha fetoprotein; OR: odds ratio value; min: minimum; max: maximum; significant value of p<0.05;

#) miRNA29c-3p as tumor suppressor;

##) miRNA21-5p and miRNA 155-5p as oncomiRs.

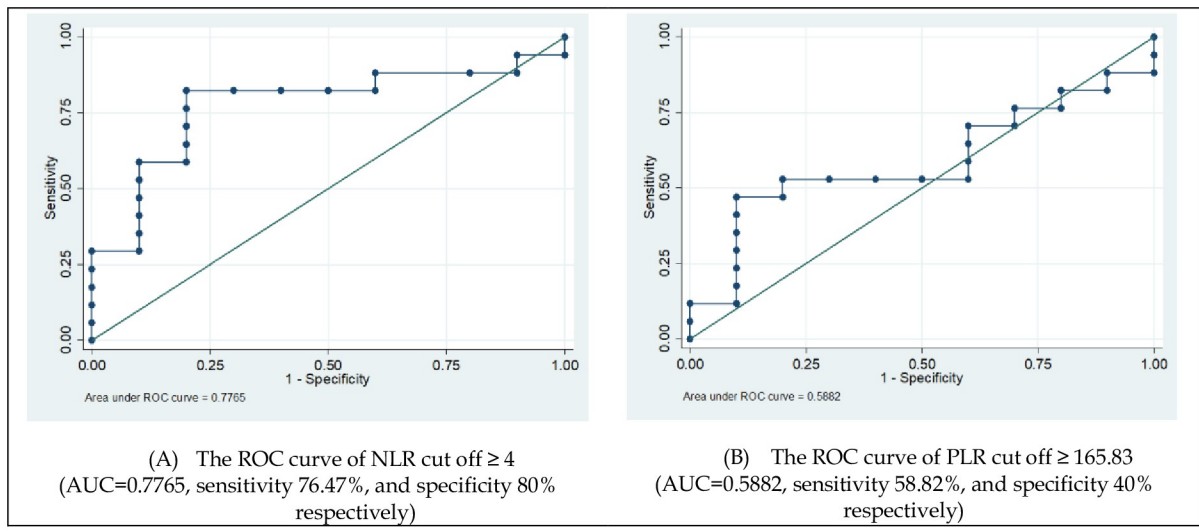

(A) The ROC curve of NLR cut off ≥ 4 (AUC=0.7765, sensitivity 76.47%, and specificity 80% respectively)

(B) The ROC curve of PLR cut off ≥ 165.83 (AUC=0.5882, sensitivity 58.82%, and specificity 40% respectively)

**Fig 1. The ROC curve of NLR cut off ≥ 4 and PLR cut off ≥ 165.83 in HCC subjects.**

(p = 0.030), actually showed in hepatitis B virus compare with other etiology significantly (p = 0.017 and p = 0.027) (Table 6).

## Survival analyses of HCC subjects

All 27 HCC subjects were followed up for survival for 3 years of monitoring. The median survival rate was 54 days, minimum of 2 days, and maximum of 851 days. The overall survival was shown in the Kaplan Meier curve in Fig 2.

The survival analyses were calculated based on every miRNA, NLR, and PLR (T1). The survival functions were shown in the Kaplan Meier curve in Fig 3A–3E. The median survival on miRNA 29c-3p subjects showed in up-regulation subjects was higher than down-regulation significantly (135 vs. 30 days, p 0.027). However, the median survival on miRNA 21-5p subject showed in up-regulation higher than down-regulation (73 vs. 45.4 days, p = 0.827), it is similarly shown in miRNA 155-5p (77.5 vs. 39 days; p = 0.393). The median survival on NLR, showed that in NLR ≥4 lower than NLR <4 significantly (29 vs. 191.5 days; p = 0.028). The median survival on PLR-T1 showed that in PLT ≥ T1 lower than PLT <T1 (46.5 vs. 321 days; p = 0.386).

Based on Hazard Ratio (HR) analyses, NLR ≥4 can predict an increasing of 1 year survival with HR 3.17 significantly (95% CI 1.285–7.830; p = 0.007). The down-regulation miRNA 29c-3p can predict a reduction of 1 year survival with HR 0.39 significantly (95% CI 0.167–0.891; p = 0.039). The HR-value for PLR-T1 was 2.12 (95% CI 0.694–6.468; p = 0.302) may predict of increasing 1 year survival, but not significant statistically. The HR value for miRNA 155-5p (HR = 0.608;95%CI 0.249–1.482; p = 0.322) and miRNA 21-5p (HR = 1.08; 95%CI 0.469–2.498; p = 0.853) can't predict the survival risk.

## Discussion

Previous studies mentioned that miRNAs might be detected clearly and are stable in serum/plasma, cancer cells, and cell lines; miRNAs are attractive as potential non-invasive biomarkers since they circulate highly in a stable, cell-free form in the blood and even saliva [5–7, 31].

**Table 6. Analyses of HCC nodule, BCLC staging and etiology with miRNAs regulation in 27 HCC subjects.**

| | miRNA 29c-3p[#] | | | miRNA 21-5p[##] | | | miRNA 155-5p[##] | | |
|---|---|---|---|---|---|---|---|---|---|
| | down-reg (9) | up-reg (18) | $p$ | down-reg (13) | up-reg (14) | $p$ | down-reg (10) | up-reg (17) | $p$ |
| Nodule count | | | | | | | | | |
| Single (8) | 2 | 6 | 0.551 | 4 | 4 | 0.901 | 4 | 4 | 0.365 |
| Multiple (19) | 7 | 12 | | 9 | 10 | | 6 | 13 | |
| Nodule diameter | | | | | | | | | |
| < 5 cm (4) | 1 | 3 | 0.702 | 1 | 3 | 0.315 | 2 | 2 | 0.561 |
| ≥ 5 cm (23) | 8 | 15 | | 12 | 11 | | 8 | 15 | |
| BCLC staging | | | | | | | | | 0.206 |
| B (8) | 3 | 5 | 0.911 | 3 | 5 | 0.767 | 3 | 5 | |
| C (15) | 5 | 10 | | 8 | 7 | | 4 | 11 | |
| D (4) | 1 | 3 | | 2 | 2 | | 3 | 1 | |
| Etiology | | | 0.362 | | | *0.009* | | | *0.030* |
| HBV (16) | 7 | 9 | 0.359[*] | 4 | 12 | 0.294[*] | 3 | 13 | 0.444[*] |
| HCV (2) | 0 | 2 | 0.326[**] | 2 | 0 | *0.017[**]* | 1 | 1 | *0.027[**]* |
| Other (9) | 2 | 7 | | 7 | 2 | | 6 | 3 | |

HCC: hepatocellular carcinoma; HBV: hepatitis B virus; HCV: hepatitis C virus; up-reg: up-regulation; down-reg: down-regulation.

BCLC: Barcelona Clinic Liver Cancer

[#]miRNA29c-3p as tumor suppressor;

[##]miRNA21-5p and miRNA 155-5p as oncomiRs;

[*] HBV vs. HCV;

[**] HBV vs. other; significant value of $p<0.05$

The regulation of miRNAs indicated that miRNAs have a role in tumor tissue dysregulation as tumor suppressors or oncogenes. The miRNAs can be used as diagnostic, targeting therapy, and prognostic markers in patients with HCC. Some of the miRNAs identified as tumor suppressors or oncogene, with the characteristic indication of poor disease progression [32, 33]. The study revealed that miRNA 29c-3p has a role in tumor-suppressing, even miRNA 21-5p, and miRNA 155-5p as tumor oncogenic. Based on Livak's Method, the regulation of miRNAs showed decreasing 1.83-fold in miRNA 29c-3p, increasing 1.74-fold in miRNA 21-5p, and increasing 1.55-fold in miRNA 155-5p, respectively. The miRNA 21 (PTEN as a target gene) and miRNA 155 (RhoA and TLR as target genes) have a role in metastasis effects of HCC. Oppositely, miRNA 29c (Bcl-2, Bcl-w, and Ras as target genes) has a role in apoptosis [33]. The circulation of miRNA 21 and miRNA 122 can predict progression LC to HCC with a value of OR >1 [34]. Plasma miRNA 21 and miRNA 122 have been considered potential biomarkers of early liver fibrosis, and they have strong correlations with fibrosis severity in HCV genotype-4. Also, they can be used for early marker detection and diagnosis of HCC if combined with plasma AFP and VEGF [10, 35]. Jin et al., 2019 revealed that plasma miRNA 122-5p was one of 5 potential biomarkers in HCC caused by chronic HBV infection [36].

Shen et al., in 2016 revealed that up-regulation of miRNA 21-5p from blood able used as an indicator for diagnosis and prognosis of HCC. The miRNA 21-5p had a role in HCC diagnosis based on: tumor suppressor and oncomiR, limitless replicative potential, liver injury, and other clinicopathologic features. The miRNA 21-5p can take a role in HCC prognosis based on: limited replicative potential and tissue invasion/metastasis [37]. And circulating miRNA 21 also can serve as a potential co-biomarker in early-stage HCC [38].

The regulation of miRNA 29 from blood and tissue, reported as an oncogene or tumor suppressor, is controversial because of various mechanisms [39]. Down-regulation of miRNA

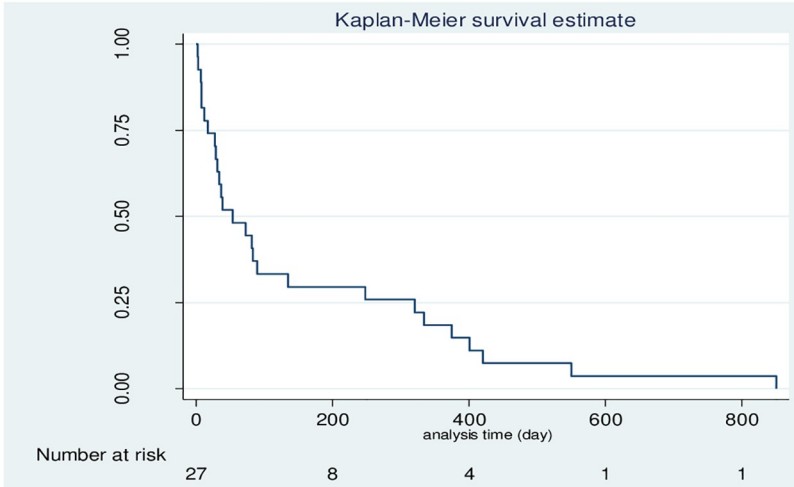

**Median survival 54 days (min 2 days; max 851 days)**

**Fig 2. Kaplan Meier survival analyses of 27 HCC patients.**

29c-3p correlated with tumor size, multiplicity pathologic features, and shorter the overall survival; and up-regulation of miRNA 29c-3p correlates with inhibition cell proliferation, apoptosis, migration, and tumor growth in vivo [40]. However, down-regulation of miRNA 29b-3p in blood correlates with tumor suppressor and oncomir only [37].

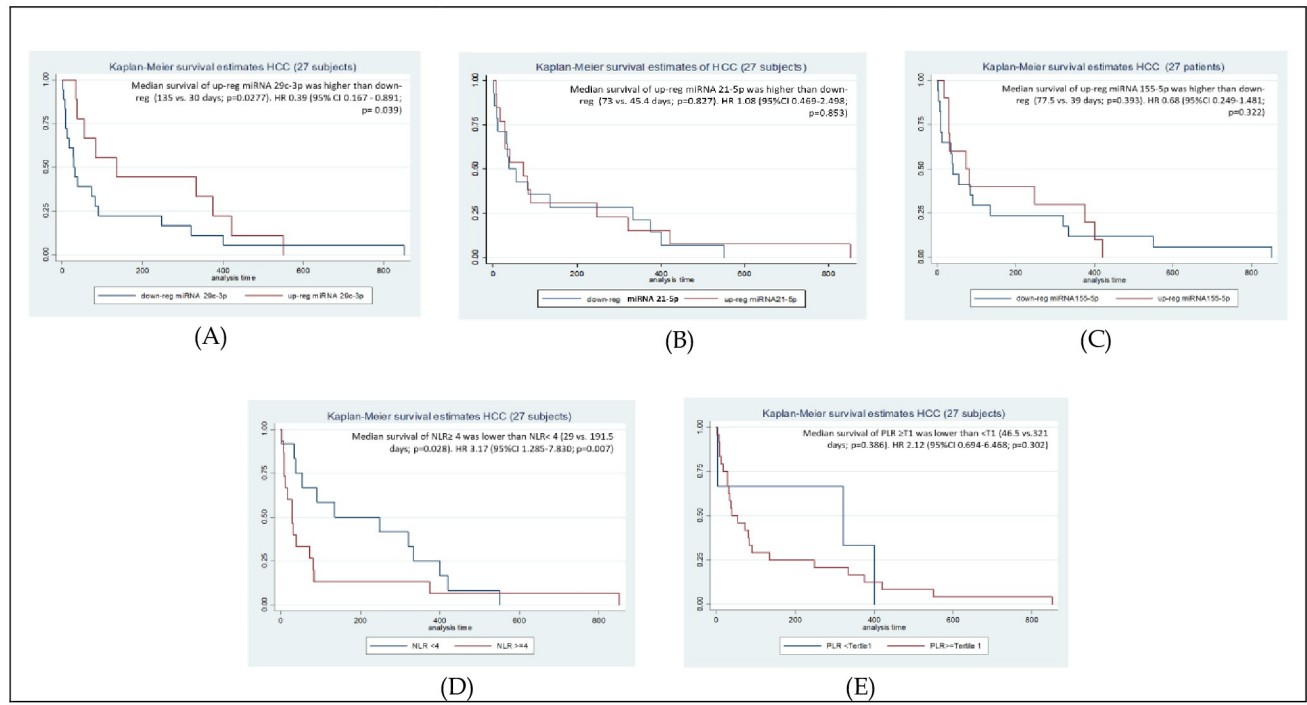

**Fig 3. Kaplan Meier survival analyses of HCC subjects based on miRNAs, NLR, and PLR(T1).** (A) Kaplan Meier survival based on miRNA 29c-3p regulation; (B) Kaplan Meier survival based on miR21-5p regulation; (C) Kaplan Meier survival based on miRNA155-5p regulation; (D) Kaplan Meier survival based on NLR cut off; and (E) Kaplan Meier survival based on PLR-Tertil 1 cut off.

Besides deregulation of miRNAs, there are discoveries of non-coding RNA (ncRNA) genes related to hepatocarcinogenesis via Wnt and STAT3 signaling pathways, although there are still limited publications until now [41].

Various miRNAs were involved in the innate and adaptive immunity in hepatic inflammation, modulated hepatic fibrosis, and activated Hepatic Stellate cells. Dysregulation of miRNAs may occur in the various cancer cells. The secreting of miRNAs from cancer cells containing exosomes, apoptotic bodies, serum proteins, and lipids. The carcinogenesis-related transcription factors may suppress some miRNAs. It is caused by presenting occurred epigenetic regulation of the other miRNAs (DNA methylation and histone modification) [42, 43].

The miRNAs expression has an effect as oncogenic or tumor-suppressive. Up-regulations miRNAs are associated with metastasis, angiogenesis, and a poor prognosis. The up-regulated miRNA 21 in HBV-HCV subjects has the possibility of being the biomarker for HCC. The up-regulated miRNA 155 with HBV also can be used as a biomarker for HCC [42, 43]. The down-regulation of miRNA 29c was associated with cell growth, apoptosis, and carcinogenesis. It was also associated with angiogenesis, migration, invasion, metastasis, and epithelial-to-mesenchymal transition [42, 43].

This study examined the relationship between three miRNAs (miRNA 29c-3p, miRNA 21-5p, and miRNA 155-5p) as potential biomarkers of HCC progression, with clinical severity based on routinely done laboratory examinations. There were significant differences in the regulation of miRNA 29c-3p based on NLR, platelet count, ChE, and albumin. Even though, in miRNA 21-5p and miRNA 155-5p, there was only a significant difference in neutrophils (see Table 4).

Based on the OR analyses, NLR$\geq$4 might be a protective factor for up-regulation of miRNA 29c-3p significantly ((HR = 0.11; p = 0.014). And as predictive factor for up-regulation of miRNA 21-5p (HR = 3; p = 0.168) and miRNA155-5p (HR = 2.63; p = 0.247) even no significant statistically (see Table 5). Previous studies mentioned that NRL $\geq$4 correlated with high mortality among low Model for End-Stage Liver Disease (MELD) scores on LC patients because the NRL correlated with pro-inflammatory neutrophils [18]. High NLR might be an independent biomarker prediction of 30-day mortality in HBV-related LC decompensated [42]. Decreasing T lymphocytes results from reducing thymic output and hyperactivation of T lymphocytes. Elevated NLR has positively correlated with pathogenesis and progression of primary liver cancer related to HBV infection [19].

PLR can be a reliable prognostic biomarker for liver cancer. Elevated platelets and reduction of lymphocytes were considered indicators of a poor prognosis and can be used to predict the overall survival of liver cancer [20, 44].

Recent studies reported that the clinical appearances might use as survival indicators such as Barcelona Clinic Liver Cancer (BCLC), portal vein thrombus, and metastases [45, 46]. Other studies focus on clinical survival and prognostic aspect based on serum and or tissue miRNA expression. The miRNA's regulation and expression take a role in the survival and prognosis of HCC.

Xie et al., 2016 revealed that the five miRNAs (miRNA 122, miRNA 126, miRNA 15a, miRNA 22, and miRNA 30a) from 847 miRNAs related to prognosis and survival in HCC patients [47]. Nagi et al., 2018 reported that five miRNAs from 173 (miRNA 584, miRNA 31, miRNA 146b-3p, miRNA 105, and miRNA 29c) showed the best performing related to prognosis value based on multivariate survival analysis [48]. The median overall survival (OS) HCC subject of the study was 54 days (range: 2–851, days) was lower than the recent study by Ali et al., 2018 (median 10.7 months; range: 9.5–12.9 months) and research by Loosen et al., 2021 (high expression was 83.24 months and 48.99 months low expression) [49, 50].

Our study result showed OS HCC subject based on NLR cut off is lower in NLR $\geq$4 than NLR<4 significantly (29 vs.191.5 days; p = 0.028), and 1 year HR 3.17 (95% CI 1.285–7.830; p = 0.007). Similarly, seen in miRNA29c-3p showed that OS up-regulation higher than down-regulation (135 vs. 30 days, p 0.027), and 1 year HR 0.39 (95% CI 0.167–0.891; p = 0.039).

Various underlying mechanisms of HCC associated with miRNAs regulation, the potential utility of miRNAs as diagnostic biomarkers, prognosis biomarkers, and innovative targeting therapy are still under investigation. However, the precise miRNAs expression for every etiology of HCC was different. This preliminary study only recruited a small sample size from two hospitals and investigated three dominant miRNAs in HCC based on recent studies.

Routine blood examinations such as NLR, PLR, AFP, albumin, and ChE have value in predicting HCC prognosis and treatment monitoring. The combination of clinical and miRNA research may improve the management of HCC targeted therapy.

This study had some limitations. (1) the observational study using the cross-sectional design for established miRNAs and followed by 3-years survival analyses; (2) the small study population conducted; (3) the blood specimen was prepared only for three potential miRNAs (2 up-regulation miRNAs as onco-miR and one down-regulation miRNA as a tumor suppressor). (3) there was no matching between groups based on age, gender, and other potential confounding factors.

## Conclusions

In conclusion, miRNA 21-5p and miRNA 155-5p might be potential biomarkers as an onco-miR in HCC subjects. The miRNA 29c-3p might be a potential biomarker as a tumor suppressor. A significant result showed that miRNA 29c-3p and NLR $\geq$4 have an opportunity as a predictive factor of clinical progression and survival.

## Acknowledgments

The authors would like to thank the members of the Internal Medicine Ward and Polyclinic Dr. Sardjito General Hospital, Internal Medicine Department Dr. Margono Soekarjo Hospital, and Cell analysis Laboratory Faculty of Medicine Public Health and Nursing Universitas Gadjah Mada.

The authors also thank the staff of Klinik Bahasa in the Office of Research and Publication of the Faculty of Medicine, Public Health and Nursing, Universitas Gadjah Mada for assistance in language editing and proofreading.

## Author Contributions

**Conceptualization:** Neneng Ratnasari, Puji Lestari, Dede Renovaldi, Juwita Raditya Ningsih, Nanda Qoriansas, Didik Setyo Heriyanto, Indwiani Astuti, Sofia Mubarika Harjana.

**Data curation:** Neneng Ratnasari, Puji Lestari, Dede Renovaldi, Juwita Raditya Ningsih, Tirta Wardana, Suharno Hakim.

**Formal analysis:** Neneng Ratnasari, Puji Lestari, Dede Renovaldi, Juwita Raditya Ningsih.

**Funding acquisition:** Sofia Mubarika Harjana.

**Investigation:** Neneng Ratnasari, Puji Lestari, Dede Renovaldi, Juwita Raditya Ningsih, Nanda Qoriansas, Suharno Hakim, Nur Signa Aini Gumilas, Fahmi Indrarti, Catharina Triwikatmani, Putut Bayupurnama.

**Methodology:** Neneng Ratnasari, Puji Lestari, Dede Renovaldi, Juwita Raditya Ningsih, Nanda Qoriansas, Tirta Wardana, Suharno Hakim, Nur Signa Aini Gumilas, Fahmi Indrarti, Catharina Triwikatmani, Putut Bayupurnama, Sofia Mubarika Harjana.

**Supervision:** Putut Bayupurnama, Didik Setyo Heriyanto, Indwiani Astuti, Sofia Mubarika Harjana.

**Validation:** Didik Setyo Heriyanto, Indwiani Astuti, Sofia Mubarika Harjana.

**Writing – original draft:** Neneng Ratnasari.

**Writing – review & editing:** Neneng Ratnasari, Puji Lestari, Dede Renovaldi, Juwita Raditya Ningsih, Tirta Wardana.

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
