## [Decision Letter · Decision Letter 0]

17 Aug 2021

PONE-D-21-23246

Potential plasma biomarkers: miRNA-29c, miRNA-21 and miRNA-155 in clinical progression of Hepatocellular Carcinoma patients

PLOS ONE

Dear Dr. Neneng Ratnasari,

Thank you for submitting your manuscript to PLOS ONE. After careful consideration, we feel that it has merit but does not fully meet PLOS ONE’s publication criteria as it currently stands. Therefore, we invite you to submit a revised version of the manuscript that addresses the points raised during the review process.

We look forward to receiving your revised manuscript.

Kind regards,

Ratna B. Ray

Academic Editor

PLOS ONE

Journal Requirements:

Reviewers' comments:

Reviewer's Responses to Questions

**Comments to the Author**

1. Is the manuscript technically sound, and do the data support the conclusions?

Reviewer #1: Yes

Reviewer #2: Partly

2. Has the statistical analysis been performed appropriately and rigorously? 

Reviewer #1: Yes

Reviewer #2: Yes

3. Have the authors made all data underlying the findings in their manuscript fully available?

Reviewer #1: Yes

Reviewer #2: Yes

4. Is the manuscript presented in an intelligible fashion and written in standard English?

Reviewer #1: Yes

Reviewer #2: No

5. Review Comments to the Author

Reviewer #1: Comments to the Author

Ratnasari et al. reported a cross-sectional study of three microRNAs (miRNA-29c, miRNA-21 and miRNA-155) and clinical factors. They showed OR analysis of the relationship between the three microRNAs and inflammation index, NLR and PLR, as well as the OR analysis of relationship between the three microRNAs and HCC tumor marker AFP. They insist miRNA-21 and miRNA-155 may act as onco-miR in HCC subjects, while miRNA-29c may act as a tumor suppressor from the results.

1. In page 2, line 6, make correction from “The collected blood samples” to “The blood samples collected from…”. None of the preceding sentences expressly provide that blood samples were collected or whom they were collected from.

2. In page 2, line 8, make correction from “and every…” to “and normalization of miRNA using…”.

3. In page 2, line 12, make correction to “who were calculated”. It is assumed “HCC patients” are not the subject of the calculation.

4. In page 2, line 17, make correction from “than healthy” to “than that of healthy control subjects”. Same for “higher than”.

5. In page 2, line 9 from the bottom, make correction from “count were showed” to “count showed”.

6. In page 10, line 5-21, add the references (9-11 and 13,14) in the correct places.

7. Please further elaborate on why you chose these three microRNAs for this investigation.

8. Is there any report about expression of these microRNAs in liver tissue or HCC?

9. In page 16, line 3 from the bottom, the sentence beginning from “Based on the……” is not very convincing. We cannot know how these microRNAs act form this examination as it is merely a cross-sectional study.

10. If you have data of overall survival, this study will be more valuable and you can add Kaplan-Meier method.

11. In page 18, line 8, “85.18% of subjects were within large tumors >5 cm in diameter and others with multiple tumors (7%). The distribution of BCLC staging C showed that more frequent than B and D.” Are there any differences between tumors numbers, tumors size or BCLC staging and these microRNAs? Are there any statistical changes? Please discuss them.

12. In page 20, line 3 from the bottom, does “AUC=0.5882” have any meaning? Please add figure of AUC.

13. In page 20, line 3 from the bottom- in page 21, line 4, it is not clear why the author included two cut-off values(i.e.165.83 and 292.40).

14. Please add statistical value in Table 5.

15. Please add details about direct targets of the three microRNAs and explanation as to miRNA-155 in the discussion part.

16. Do you have any data between clinical characteristics and matching these microRNAs?

17. How will this report be helpful for medical treatment of LC or hepatitis patients? Please discuss.

Reviewer #2: In this study, Ratnasari et al. evaluated the expression level of miRNA-29c, miRNA-21, and miRNA-155 in plasma of 27 HCC confirmed patients and compared them with healthy controls. The correlation study with miRNA levels and clinical progression parameters concluded that miR-29c and miR-155 could be used as potential biomarkers in HCC patients. Overall, the study presents limited information and defines HCC in general without dissecting the effect of causing agents on miRNA expression. Among the 27 HCC patients, 16 HBV and 2 were HCV-positive patients. However, it is not clear whether there are any differences at the level of the miRNA within the studied groups. Moreover, the overall sample size is not adequate to conclude as a biomarker. Even within the different age groups, it is not clear how these miRNAs vary.

Fold change is expressed as a relative expression against miR-16. However, an actual copy number is required to define miRNAs levels as a biomarker. The basal expression level for miR-16 within healthy and HCC samples need to be shown.

Overall, this study reflects a general correlation study, and the conclusion part is simply overstated.

6. PLOS authors have the option to publish the peer review history of their article (what does this mean?). If published, this will include your full peer review and any attached files.

Reviewer #1: No

Reviewer #2: No

---

## [Author Response · Author response to Decision Letter 0]

5 Jan 2022

Thank you vary much for all comments to our manuscript. We hope the manuscript after re-revision may answer all reviewer's question and suggestion. We hope our revision manuscript as a final article. And the final article may improved our next basic to clinic research especially in HCC field.

---

## [Decision Letter · Decision Letter 1]

18 Jan 2022

Potential plasma biomarkers: miRNA-29c, miRNA-21 and miRNA-155 in clinical progression of Hepatocellular Carcinoma patients

PONE-D-21-23246R1

Dear Dr. Neneng Ratnasari,

We’re pleased to inform you that your manuscript has been judged scientifically suitable for publication and will be formally accepted for publication once it meets all outstanding technical requirements.

Kind regards,

Ratna B. Ray

Academic Editor

PLOS ONE

Additional Editor Comments (optional):

Reviewers' comments:

Reviewer's Responses to Questions

**Comments to the Author**

1. If the authors have adequately addressed your comments raised in a previous round of review and you feel that this manuscript is now acceptable for publication, you may indicate that here to bypass the “Comments to the Author” section, enter your conflict of interest statement in the “Confidential to Editor” section, and submit your "Accept" recommendation.

Reviewer #2: All comments have been addressed

2. Is the manuscript technically sound, and do the data support the conclusions?

Reviewer #2: Yes

3. Has the statistical analysis been performed appropriately and rigorously? 

Reviewer #2: Yes

4. Have the authors made all data underlying the findings in their manuscript fully available?

Reviewer #2: Yes

5. Is the manuscript presented in an intelligible fashion and written in standard English?

Reviewer #2: Yes

6. Review Comments to the Author

Reviewer #2: The revised version of the manuscript has significantly improved in terms of data presentation, statistical analysis, and correlation studies with different clinical parameters. However, the title of the manuscript may be revised. Since the sample size has a limitation, instead of emphasizing 'Potential biomarker,' it may revise as biomarker potential in the title.

7. PLOS authors have the option to publish the peer review history of their article (what does this mean?). If published, this will include your full peer review and any attached files.

Reviewer #2: No

---

## [Editor Report · Acceptance letter]

3 Feb 2022

PONE-D-21-23246R1 

Potential plasma biomarkers: miRNA-29c, miRNA-21, and miRNA-155 in clinical progression of Hepatocellular Carcinoma patients 

Dear Dr. Ratnasari:

I'm pleased to inform you that your manuscript has been deemed suitable for publication in PLOS ONE. Congratulations! Your manuscript is now with our production department. 

Kind regards, 

on behalf of

Dr. Ratna B. Ray 

Academic Editor

PLOS ONE